# The Neutrophil-to-Lymphocyte and Platelet-to-Lymphocyte Ratios Predict Reperfusion and Prognosis after Endovascular Treatment of Acute Ischemic Stroke

**DOI:** 10.3390/jpm11080696

**Published:** 2021-07-22

**Authors:** Sang-Hwa Lee, Min Uk Jang, Yerim Kim, So Young Park, Chulho Kim, Yeo Jin Kim, Jong-Hee Sohn

**Affiliations:** 1Department of Neurology, Chuncheon Sacred Heart Hospital, Hallym University College of Medicine, Chuncheon 24253, Korea; bleulsh@naver.com (S.-H.L.); gumdol52@naver.com (C.K.); yjhelena@hanmail.net (Y.J.K.); 2Institute of New Frontier Research Team, Hallym University, Chucheon 24253, Korea; 3Department of Neurology, Dongtan Sacred Heart Hospital, Hallym University College of Medicine, Hwaseong 18450, Korea; mujang@gmail.com; 4Department of Neurology, Kangdong Sacred Heart Hospital, Hallym University College of Medicine, Seoul 05355, Korea; brainyrk@gmail.com; 5Department of Endocrinology and Metabolism, Kyung Hee University Hospital, Seoul 02447, Korea; malcoy@hanmail.net

**Keywords:** neutrophil/lymphocyte ratio, platelet/lymphocyte ratio, endovascular treatment, reperfusion rate, infarct volume

## Abstract

Background: Studies assessing the prognostic effect of inflammatory markers of blood cells on the outcomes of patients with acute ischemic stroke treated with endovascular treatment (EVT) are sparse. We evaluated whether the neutrophil-to-lymphocyte ratio (NLR) and platelet-to-lymphocyte ratio (PLR) affect reperfusion status in patients receiving EVT. Methods: Using a multicenter registry database, 282 patients treated with EVT were enrolled in this study. The primary outcome measure was unsuccessful reperfusion rate after EVT defined by thrombolysis in cerebral infarction grades 0–2a. Logistic regression analysis was performed to analyze the association between NLR/PLR and unsuccessful reperfusion rate after EVT. Results: Both NLR and PLR were higher in the unsuccessful reperfusion group than in the successful reperfusion group (*p* < 0.001). Multivariate analysis showed that both NLR and PLR were significantly associated with unsuccessful reperfusion (adjusted odds ratio (95% confidence interval): 1.11 (1.04–1.19), PLR: 1.004 (1.001–1.01)). The receiver operating characteristic curve showed that the predictive ability of both NLR and PLR was close to good (area under the curve (AUC) of NLR: 0.63, 95% CI (0.54–0.72), *p* < 0.001; AUC of PLR: 0.65, 95% CI (0.57–0.73), *p* < 0.001). The cutoff values of NLR and PLR were 6.2 and 103.6 for unsuccessful reperfusion, respectively. Conclusion: Higher NLR and PLR were associated with unsuccessful reperfusion after EVT. The combined application of both biomarkers could be useful for predicting outcomes after EVT.

## 1. Introduction

Since 2015, guidelines have indicated that endovascular treatment (EVT) plays an important role in determining the prognosis of acute ischemic stroke [1,2]. The rate of successful reperfusion after EVT has been reported to range from 41% to 88% in several major clinical trials [3]. Since reperfusion is known to be a key factor in determining the outcome of patients with acute ischemic stroke [4], evaluating markers to predict reperfusion status after EVT could allow clinicians to raise interest in real-world practice of EVT.

Inflammatory response is one of the major pathophysiological mechanisms that cause atherosclerosis, and several circulatory inflammatory markers are known to be commonly activated in acute cerebral infarction [5,6,7,8,9]. Activated inflammatory reactions affect the ischemic area and generate several destructive materials (reactive oxygen species, proteases, matrix metalloproteinase-9, cytokines), contributing to the exacerbation of cerebral infarction [10,11,12,13]. Hence, several studies have shown that these inflammatory reactions promote worsening symptoms and poor prognosis after acute ischemic stroke. Determining the neutrophil-to-lymphocyte ratio (NLR) and platelet-to-lymphocyte ratio (PLR) is a rapid and easy method used to determine this inflammatory response [14,15,16]. Novel biomarkers that reflect the baseline inflammatory process in patients with ischemic stroke are widely used in studies to predict the prognosis of several subjects with cerebrovascular disease [16,17,18,19].

Although several studies have investigated how these inflammatory markers affect the outcomes of patients receiving EVT [12,20,21,22], no studies have investigated how these inflammatory markers affect the reperfusion rate after EVT. In this study, we aimed to evaluate the association between NLR/PLR and reperfusion rate and final infarct volume after EVT using a multicenter database, thereby exploring the practical utility of these markers in predicting post-EVT outcomes.

## 2. Methods

### 2.1. Study Population

We consecutively registered all patients with acute ischemic stroke between March 2015 and January 2021 in two university-affiliated institutions (Chuncheon Sacred Heart Hospital and Dongtan Sacred Heart Hospital). All enrolled patients received acute stroke management according to the institutional protocols based on recent guidelines. We identified patients with acute ischemic stroke who were treated with EVT. In this study, we excluded the following patients: (1) patients not undergoing follow-up brain computed tomography (CT) or magnetic resonance imaging within 24 h of stroke onset; (2) patients with no available data on neutrophil, lymphocyte, and platelet count; (3) patients not undergoing a perfusion study (e.g., multiphasic computed tomography angiography (mCTA)); (4) patients with Alberta Stroke Program Early CT Score < 6; and (5) patients with a pre-stroke modified Rankin Scale (mRS) score ≥ 2.

### 2.2. Data Collection and Definition of Parameters

We obtained the following data directly from the registry database: (1) demographics, including age and sex; (2) stroke risk factors, medical history, prior stroke, hypertension, diabetes mellitus, hyperlipidemia, atrial fibrillation, current smoking status, pre-stroke status, and prior use of statin and antithrombotic drugs; (3) stroke characteristics, acute stroke treatment, initial National Institute of Health Stroke Scale score, ischemic stroke mechanism according to the Trial of Org 10172 in Acute Stroke Treatment classification with some modifications [23], tissue plasminogen activator dose, and reperfusion therapy (intravenous thrombolysis (IVT) and EVT); and (4) laboratory data, including hemoglobin, white blood cell count, neutrophil count, platelet count, lymphocyte count, serum creatinine, initial random glucose, fasting low-density lipoprotein, prothrombin time, glycated hemoglobin (HbA1c), and systolic blood pressure.

All complete blood cell samples were collected at the time of hospitalization before initiating EVT. Complete blood cell samples were immediately centrifuged (2000 rpm for 20 min at 4 °C) after being collected in a calcium ethylenediaminetetraacetic acid tube. Subsequently, the cell counts were analyzed using the same auto-analyzer (XE-2100, Sysmex, Kobe, Japan) in two institutions. The NLR was calculated by dividing the number of neutrophils by the number of lymphocytes. The PLR was calculated by dividing the number of platelets by the number of lymphocytes.

In the present study, we analyzed pre-EVT mCTA to evaluate baseline leptomeningeal collateral status. We followed the imaging protocols of the Calgary Stroke Program of the University of Calgary and classified the collateral status into three clinically relevant groups [24]. Collateral grade was categorized as good (grade 4 or 5), intermediate (grade 2 or 3), and poor (grade 0 or 1). Collateral status was quantified by two experienced vascular neurologists (S.-H.L. and M.U.J.) in a double-blind manner. The inter-rater reliability for the evaluation of collateral status was excellent (intraclass correlation coefficient (ICC) 0.91, *p* < 0.001).

The primary outcome measure was poor reperfusion rate defined by modified thrombolysis in cerebral infarction (TICI) grades 0–2a [25]. Vascular neurologists in each institution made decisions on whether to perform reperfusion therapy. Regarding EVT procedures, the choice of device and intervention strategy was made at the discretion of the interventionists at each institution, according to recent guidelines. The secondary outcome measure was the final infarct volume after EVT and mRS at 3 months. The infarct volume confirmed by diffusion-weighted imaging was calculated using Medical Image Processing and Visualization software (version 7.3.0, National Institutes of Health, Bethesda, MD, USA). The infarct volume was quantified by two experienced vascular neurologists (S.-H.L. and M.U.J.) in a double-blind manner. The inter-rater reliability for the evaluation of the infarct volume was good (ICC 0.85, *p* < 0.001).

### 2.3. Statistical Analyses

We hypothesized that a high NLR and PLR could be associated with an increased poor reperfusion rate and final infarct volume after EVT. Summary statistics are presented as the number of subjects (percentage) for categorical variables and as mean ± standard deviation or median (interquartile range) for continuous variables. Group comparisons were made using Pearson’s chi-squared test for categorical variables and Student’s t-test or the Mann–Whitney U test for continuous variables, as appropriate.

Regarding the primary and secondary outcome measures, the higher NRL and PLR were compared using Pearson’s chi-squared test for categorical variables and Student’s t-test or the Mann–Whitney U test for continuous variables. To evaluate the independent effects of dichotomized NRL and PLR on outcome measures, we performed a binary logistic regression analysis. Variables for adjustment in the multivariate analysis were selected if their *p* values were <0.1 in comparison according to the NRL and PLR and if their associations with each outcome variable were clinically plausible. Crude and adjusted odds ratios (ORs) and 95% confidence intervals (CIs) were estimated.

To assess the predictive ability of NLR and PLR on unsuccessful reperfusion, we constructed a receiver operating characteristic (ROC) curve using the “pROC” package of R. The 95% CI for area under the curve (AUC) and *p* value were calculated using Delong’s test. The cutoff values of NLR and PLR for poor reperfusion were calculated using the Youden index.

A sensitivity analysis using linear regression was performed to evaluate whether both NLR and PLR could be associated with infarct volume in anterior circulation infarctions only.

Subgroup analysis was performed to evaluate the impact of NLR and PLR on unsuccessful reperfusion according to stroke mechanisms. We separately performed binary logistic regression analysis in patients with stroke with large artery atherosclerosis (LAA) and cardioembolism (CE) using the aforementioned statistical methods.

Statistical analyses were performed using the International Business Machines (IBM) Statistical Package for the Social Sciences version 21.0 software (IBM Corporation, Armonk, NY, USA) and R version 4.0.3 (R Core Team 2020; R Foundation for Statistical Computing, Vienna, Austria).

## 3. Results

We analyzed 3169 consecutive patients with acute ischemic stroke. Among these patients, 606 underwent IVT and EVT. Of the 606 patients, 282 who underwent EVT were included in our study. Of the 282 patients, 58 (20.6%) were in the unsuccessful reperfusion group (Appendix A). The unsuccessful reperfusion group likely had fewer prior statin users than the successful reperfusion group. Some laboratory results (white blood cell count, low-density lipoprotein level, and HbA1c level) showed statistically but not clinically significant differences between the two groups (Table 1).

Both NLR and PLR were higher in the unsuccessful reperfusion group than in the successful reperfusion group (*p* < 0.001, Table 2). In addition, infarct volume and poor functional outcome at 3 months (mRS > 2) were higher in the unsuccessful reperfusion group than in the successful reperfusion group. NLR and PLR showed increasing trends with increasing infarct volume tertile (p for trend = 0.04, Figure 1).

Multivariate analysis showed that both NLR and PLR were significantly associated with unsuccessful reperfusion (adjusted OR (95% CI): 1.11 (1.04–1.19), PLR: 1.004 (1.001–1.01)). In addition, the highest tertile of NLR and PLR was also associated with unsuccessful reperfusion (Table 3 and Appendix A). With respect to poor functional outcome at 3 months (mRS > 2), both NLR and PLR were independently associated with poor functional outcome (adjusted OR (95% CI): 1.11 (1.01–1.19), PLR: 1.004 (1.001–1.01), Table 4, Appendix A). Both the highest tertile of NLR and PLR also increased the poor functional outcome.

In linear regression analysis, NLR and PLR were significantly associated with increased infarct volume (*p* = 0.004, r = 0.41, r^2^ = 0.17 in NLR, *p* = 0.02, r = 0.40, r^2^ = 0.16 in PLR). As a sensitivity analysis of the dataset of subjects with only anterior circulation infarct, NLR was associated with increased infarct volume, but PLR was not (Appendix A).

The ROC curve showed that the predictive ability of both NLR and PLR was close to good (AUC of NLR: 0.63, 95% CI (0.54–0.72), *p* < 0.001; AUC of PLR: 0.65, 95% CI (0.57–0.73), *p* < 0.001). There were no significant differences in the prediction of unsuccessful reperfusion between NLR and PLR. The cutoff values of NLR and PLR were 6.2 and 103.6 for unsuccessful reperfusion, respectively (Figure 2).

In subgroup analysis, both NLR and PLR were significantly associated with unsuccessful reperfusion in patients with LAA (adjusted OR (95% CI): NLR 1.35 (1.05–1.75), PLR 1.01 (1.002–1.02), Appendix A). Interestingly, only NLR was associated with unsuccessful reperfusion in patients with CE, but the impact of PLR was attenuated (adjusted OR (95% CI): NLR 1.17 (1.04–1.32), PLR 1.004 (0.999–1.01), Appendix A).

## 4. Discussion

The main findings of this study are as follows: (1) both higher NLR and PLR had higher unsuccessful reperfusion rates and higher infarct volumes, (2) both NLR and PLR could be strong predictors for unsuccessful reperfusion and poor functional outcome, and (3) the predictive ability of NLR and PLR for unsuccessful reperfusion could be reliable with cutoff values of 6.2 and 103.6, respectively.

After vessel occlusion in acute stroke, the inflammatory cascade is activated on the vessel endothelium and platelets immediately [26,27]. The intravascular endothelium and platelet are stressed by stagnant blood flow and induced release of P-selectin in the endothelium and platelet. P-selectin could bind to circulating leukocytes, facilitating cluster formation of leukocytes, causing intravascular clogging on the endothelial surface. Neutrophils and platelets play a key role in the inflammation and thrombosis occurring subsequent to inflammatory cascades. The NLR and PLR may be novel indicators of the intensity of systemic inflammation, which may allow quantification of these inflammatory reactions. Higher PLR in particular has been associated with thrombus and inflamed intravascular plaque formation. In addition, activated platelets contribute to plaque destabilization by promoting thrombi and inducing an inflammatory response [28]. Therefore, we can speculate that higher NLR and PLR may be associated with an increased risk of unsuccessful reperfusion. Although this intravascular inflammatory reaction occurs, inflammation also initiates the breakdown of the blood–brain barrier (BBB), and cytokines and adhesion molecules invade the brain parenchyma, thereby further enhancing ischemic damage [29,30,31]. Several evidences have shown that neutrophils, which are activated early in intravascular adhesion, contribute to postischemic injury by intravascular clogging, destabilizing BBB by matrix proteinases, and generating reactive oxygen species [32,33]. This pathological mechanism underlying the inflammatory reaction after acute ischemic stroke may explain the potential associations between high NLR and PLR and both reperfusion rate and infarct volume after EVT. Accordingly, these pathological findings support our main results.

The application of NLR and PLR as rapidly tested and inexpensive biomarkers for systemic inflammation proved to be useful in diagnosing and predicting cerebrovascular disease. The ratios were more stable indices, with a single blood parameter influenced by several physiological and pathological conditions. In addition, the combined use of NLR and PLR reflected both pro-inflammatory and procoagulant status before EVT, helping to precisely select biomarkers. Hence, the combined application of both NLR and PLR was rational for predicting outcomes after EVT. Previous studies that limited subjects receiving EVT have shown that NLR and PLR could be predictors for hemorrhagic transformation, functional outcome, and mortality [12,21,22,34]. Our novel findings also showed that the different predictive values of NLR and PLR for reperfusion state broadened the choice of biomarkers to establish the EVT strategy. We tentatively propose that, although there are other strong predictors for successful reperfusion, joint consideration of NLR/PLR may help predict reperfusion status by blood sampling early after admission.

Our secondary outcomes were partly consistent with those of a previous study of NLR and PLR. NLR and PLR are associated with short-term outcome, severity, and mortality of ischemic stroke [10,13,15,17]. Recently, NLR and PLR were associated with hemorrhagic transformation and outcomes in patients treated with IVT [16,18,19]. The association between both biomarkers and hemorrhagic transformation after IVT was explained by BBB breakdown and increasing permeability via inflammatory reaction in previous studies. Our findings on the association between NLR/PLR and final infarct volume size after EVT could share similar pathophysiologic processes. Interestingly, our study showed that PLR was not associated with an increased final infarct volume in subjects with anterior circulation infarction. In contrast with our results, a previous study with a small sample size (*n* = 57) has shown that crude high PLR values have increased infarct cores [22]. Since the final infarct size could be markedly influenced by several other individual and institutional variables (collateral status, interval from onset to puncture time) [35], although these variables were not statistically different between the successful and unsuccessful reperfusion groups in our database (data not shown), our results could be attenuated. In addition, PLR plays a key role in thrombus and inflamed plaque formation, especially under atherosclerotic conditions [36]. The majority of our patients with anterior circulation infarction were found to have CE (LAA: 28.5% and CE: 55.0%). The low rate of LAA could explain the attenuated association between PLR and infarct volume in anterior circulation infarction. Further studies with large sample sizes and advanced programs for precise calculation of infarct size are warranted to generalize our results.

Notably, the effects of NLR and PLR on reperfusion state seem to differ according to the stroke mechanism in our study. Both NLR and PLR were more valuable in predicting the reperfusion state after EVT in acute ischemic stroke with LAA than in those with CE. Previous studies have demonstrated that NLR and PLR are associated with atherosclerotic development and atherosclerotic plaque destabilization [37,38]. In addition, histological studies have revealed that neutrophils were prone to be found in ruptured atherosclerotic plaque [39]. Furthermore, low levels of lymphocytes could be associated with atherosclerosis because of the lack of neuroprotective effect of CD4 T cells [31]. Since the inflammatory reaction is initiated on the endothelial surface, we carefully assumed that the impact of NLR and PLR on the reperfusion state could be more robust in LAA than CE, which originated from intracardiac pathology. We could not generalize this result due to the small sample size of each stroke mechanism. Further studies are warranted to address this issue.

Despite the strengths of this study, including its two-consecutive registry database, this study has several limitations. First, all data with a moderate sample size were retrospectively collected from the registry database. Second, since NLR and PLR were sampled once without serial measurements, we could not reveal the association between these indices and outcomes after EVT. However, since the main aim of this study was to demonstrate the impact of NLR and PLR on reperfusion at the acute stage, dynamic changes could be inevitable in our study. Third, although we adjusted for several variables that may affect the outcomes, unmeasured confounding factors could hinder the applicability of our main findings. The outcomes after EVT can be influenced by procedure-related factors, including the type of stent retriever, operator technique and expertise, and procedure time, which were not available for analysis in our study. Fourth, although the aim of this study was to evaluate the impact of NLR and PLR on outcome, other inflammatory markers that arise during the acute stroke stage were not available in this study.

## 5. Conclusions

We suggest that both NLR and PLR, tested easily and rapidly at the acute stroke stage, were independently associated with unsuccessful reperfusion and poor outcome after EVT. Joint application of both NLR and PLR could be useful and reliable for predicting the reperfusion state after EVT, especially in patients with LAA stroke. However, further studies are required to address the practical application of NLR and PLR in clinical settings.

## Figures and Tables

**Figure 1 jpm-11-00696-f001:**
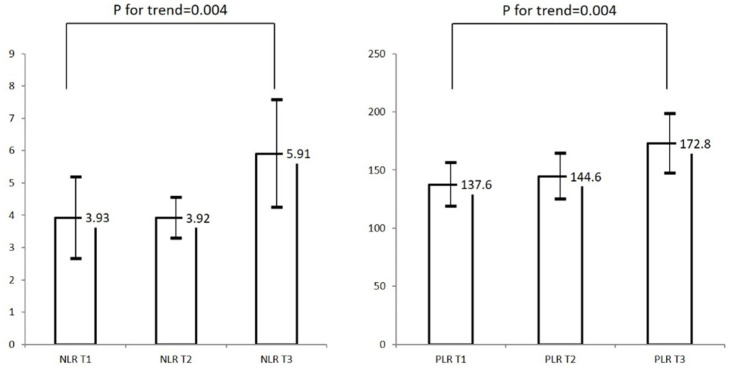
Distribution of final infarct volume according to tertile of NLR and PLR. Abbreviations: NLR, neutrophil-to-lymphocyte ratio; PLR, platelet-to-lymphocyte ratio; T1, lowest tertile; T2, middle tertile; T3, highest tertile. Range of tertiles: NLR T1 (0.30–1.88), NLR T2 (1.90–4.15), NLR T3 (4.15–54.75); PLR T1 (24.5–94.1), PLR T2 (94.2–157.2), PLR T3 (158.6–754.4).

**Figure 2 jpm-11-00696-f002:**
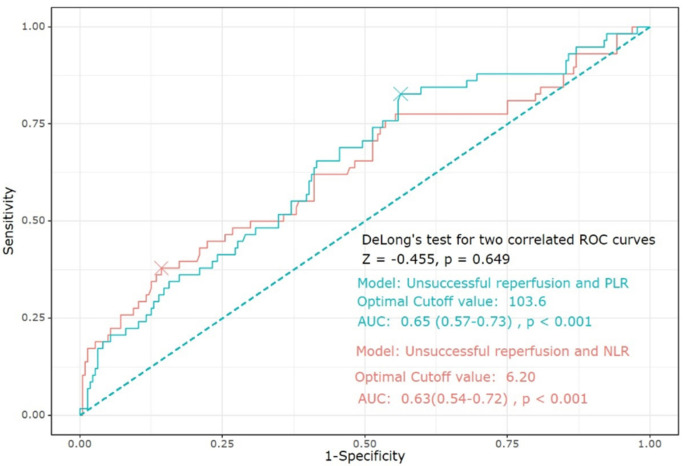
The ROC curve showing the predictive ability of both NLR and PLR for unsuccessful reperfusion. Abbreviations: ROC, receiver operating characteristic; NLR, neutrophil-to-lymphocyte ratio; PLR, platelet-to-lymphocyte ratio.

**Table 1 jpm-11-00696-t001:** Baseline characteristics according to reperfusion status.

	Successful Reperfusion (*n* = 224)	Unsuccessful Reperfusion (*n* = 58)	*p*-Value
Age (SD)	68.7 (13.7)	72.8 (11.7)	0.17
Male, % (SD)	135 (60.3)	27 (46.6)	0.07 *
BMI (SD)	25.5 (23.2)	23.4 (3.8)	0.59 ^†^
NIHSS (IQR)	14 (9–18)	15 (11-18)	0.13 ^‡^
Interval from arrival to puncture, min (IQR)	96.5 (75.0–134.0)	101.5 (73.0–156.0)	0.18 ^‡^
Stroke mechanism, % (SD)			0.12 *
LAA	55 (24.6)	21 (36.2)	
CE	130 (58.0)	25 (43.1)	
Others	39 (17.4)	12 (20.7)	
Previous stroke, % (SD)	49 (21.9)	10 (17.2)	0.44 *
HTN, % (SD)	121 (54.0)	38 (65.5)	0.12 *
DM, % (SD)	62 (27.7)	15 (25.9)	0.78 *
HL, % (SD)	33 (14.7)	7 (12.1)	0.60 *
Current smoking, % (SD)	31 (13.8)	7 (12.1)	0.73 *
Atrial fibrillation, % (SD)	120 (53.6)	25 (43.1)	0.16 *
Previous use of antithrombotics, % (SD)	87 (38.8)	18 (31.0)	0.27 *
Previous use of statin, % (SD)	49 (21.9)	5 (8.6)	0.02 *
WBC, x1000/µL (SD)	8.70 (3.26)	9.46 (4.42)	0.03 ^†^
Creatinine, mg/dL (SD)	0.98 (0.60)	0.87 (0.30)	0.36 ^†^
Hemoglobin, g/dL (SD)	13.7 (2.1)	13.4 (2.2)	0.76 ^†^
Platelet, x1000/µL (SD)	221.9 (70.9)	250.9 (125.9)	0.02 ^†^
LDLc, mg/dL (SD)	46.1 (30.8)	48.2 (34.8)	0.03 ^†^
HbA1c, % (SD)	6.2 (1.4)	5.9 (0.7)	<0.001 ^†^
Prothrombin time, INR (SD)	1.08 (0.25)	1.05 (0.11)	0.22 ^†^
CRP, mg/dL (SD)	12.2 (25.0)	14.0 (26.2)	0.28 ^†^
Initial random glucose, mg/dL (SD)	141.2 (51.6)	148.9 (63.7)	0.95 ^†^
SBP, mmHg (SD)	149.2 (26.4)	149.3 (25.9)	0.70 ^†^

Abbreviations: SD, standard deviation; BMI, body mass index; NIHSS, National Institute Health of Stroke Scale; IQR, interquartile range; LAA, large artery atherosclerosis; CE, cardioembolism; HTN, hypertension; DM, diabetes mellitus; HL, hyperlipidemia; EVT, endovascular treatment; IVT, intravenous thrombolysis; WBC, white blood cell; LDLc, low-density lipoprotein cholesterol; HbA1c, glycated hemoglobin; INR, international normalized ratio; CRP, C-reactive protein; SBP, systolic blood pressure. * Calculated using the chi-square test. ^†^ Calculated using Student’s *t*-test. ^‡^ Calculated using the Mann–Whitney U test.

**Table 2 jpm-11-00696-t002:** Intervention-related findings and outcomes according to reperfusion status.

	Successful Reperfusion (*n* = 224)	Unsuccessful Reperfusion (*n* = 58)	*p*-Value
Reperfusion therapy, % (SD)			0.67 *
EVT only	105 (46.9)	29 (50.0)	
Combined IVT and EVT	119 (53.1)	29 (50.0)	
LVO location, % (SD)			0.76 *
ICA	17 (7.6)	3 (5.2)	
M1/M2 MCA	183 (81.7)	46 (79.3)	
BA/VA	24 (10.7)	9 (15.5)	
Collateral status, % (SD)			0.06 *
Poor	100 (44.6)	22 (37.9)	
Intermediate	35 (15.6)	17 (29.3)	
Good	89 (39.7)	19 (32.8)	
NLR (IQR)	2.48 (1.52–4.87)	3.92 (2.29–8.14)	0.002 ^†^
PLR (IQR)	113.2 (80.9–168.8)	147.2 (109.0–227.9)	0.001 ^†^
Infarct volume, cm^3^ (IQR)	12.25 (2.64–50.78)	28.88 (9.20–102.63)	<0.001 ^†^

Abbreviations: SD, standard deviation; EVT, endovascular treatment; IVT, intravenous thrombolysis; LVO, larger vessel occlusion; ICA, internal carotid artery; MCA, middle cerebral artery; BA, basilar artery; VA, vertebral artery; NLR, neutrophil/lymphocyte ratio; PLR, platelet/lymphocyte ratio. * Calculated using the chi-square test. ^†^ Calculated using the Mann–Whitney U test.

**Table 3 jpm-11-00696-t003:** Logistic regression analysis showing effect of NLR and PLR on unsuccessful reperfusion.

	aOR	95% CI	*p*-Value		aOR	95% CI	*p*-Value
raw NLR	1.11	1.04–1.19	0.001	raw PLR	1.004	1.001–1.01	0.01
NLR T1	reference			PLR T1	reference		
NLR T2	1.33	0.56–3.12	0.52	PLR T2	2.56	1.06–6.22	0.04
NLR T3	2.51	1.11–5.70	0.03	PLR T3	2.84	1.15–6.99	0.02

Abbreviations: NLR, neutrophil/lymphocyte ratio; PLR, platelet/lymphocyte ratio; aOR, adjusted odds ratio; CI, confidence interval; T1, lowest tertile; T2, middle tertile; T3, highest tertile. Range of tertiles: NLR T1 (0.30–1.88), NLR T2 (1.90–4.15), NLR T3 (4.15–54.75); PLR T1 (24.5–94.1), PLR T2 (94.2–157.2), PLR T3 (158.6–754.4).

**Table 4 jpm-11-00696-t004:** Logistic regression analysis showing effect of NLR and PLR on poor functional outcome (mRS 3–6).

	aOR	95% CI	*p*-Value		aOR	95% CI	*p*-Value
raw NLR	1.20	1.06–1.35	0.004	raw PLR	1.01	1.004–1.02	<0.001
NLR T1	reference			PLR T1	reference		
NLR T2	2.32	1.15–4.70	0.02	PLR T2	1.52	0.75–3.09	0.24
NLR T3	3.67	1.67–8.06	0.001	PLR T3	2.59	1.10–5.55	0.02

Abbreviations: NLR, neutrophil/lymphocyte ratio; PLR, platelet/lymphocyte ratio; mRS; modified Rankin Scale; aOR, adjusted odds ratio; CI, confidence interval; T1, lowest tertile; T2, middle tertile; T3, highest tertile. Range of tertiles: NLR T1 (0.30–1.88), NLR T2 (1.90–4.15), NLR T3 (4.15–54.75); PLR T1 (24.5–94.1), PLR T2 (94.2–157.2), PLR T3 (158.6–754.4).

## Data Availability

All data generated or analyzed during this study are included in this published article. Anonymized data will be shared by request from any qualified investigator.

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
