# Peer review of "The Neutrophil-to-Lymphocyte and Platelet-to-Lymphocyte Ratios Predict Reperfusion and Prognosis after Endovascular Treatment of Acute Ischemic Stroke"

_jpm, 2021, doi:10.3390/jpm11080696_

Round 1
Reviewer 1 Report
- Table number 1– is too long with many demographic, laboratory, interventional and outcome variables. It needs to be divided into several tables when one table will include the interventional findings, NLR, PLR and outcome.
- Results – line 145-147: “The unsuccessful reperfusion group likely had lower pre-stroke statin level, higher white blood cell count and low-density lipoprotein level, and lower HbA1c level than the successful reperfusion group (Table 1).” The statistically difference between the group are they have any clinical relevance? The numbers are actually in the normal range??
- In the unsuccessful reperfusion group had a significant increase of infarct volume compare to the successful group. Maybe this finding can explain the effect of the inflammatory response and the high NLR in this group?
- The authors suggest (line 228): “In the trend of increasing performance of EVT according to recent stroke guidelines, our study could raise the interest of application of NLR and PLR to predict outcomes in real-world practice of EVT”. It is unclear how physicians can use these finding to inform patient about his prognosis and who this finding will change the clinical practice for the individual patient.
- A more detailed explanation is needed for this finding “PLR was not associated with an increased final infarct volume in subjects with anterior circulation infarction”??
- It is a two center retrospective study and not a multi-center study (line 256)
Author Response
Response to reviewers
Reviewer #1
1.Table number 1– is too long with many demographic, laboratory, interventional and outcome variables. It needs to be divided into several tables when one table will include the interventional findings, NLR, PLR and outcome.
Response: We appreciate your comment. As per the Reviewer’s suggestion, we have divided Table 1 into two tables, with one showing patient demographics and laboratory findings, and the other showing the interventional findings and outcomes, as shown below.
[Table]
Table 1. Baseline characteristics according to reperfusion status
Successful reperfusion (n=224) |
Unsuccessful reperfusion (n=58) |
p-value |
|
Age (SD) |
68.7 (13.7) |
72.8 (11.7) |
0.17 |
Male, % (SD) |
135 (60.3) |
27 (46.6) |
0.07* |
BMI (SD) |
25.5 (23.2) |
23.4 (3.8) |
0.59† |
NIHSS (IQR) |
14 (9-18) |
15 (11-18) |
0.13‡ |
Interval from arrival to puncture, min (IQR) |
96.5 (75.0-134.0) |
101.5 (73.0-156.0) |
0.18‡ |
Stroke mechanism, % (SD) |
0.12* |
||
LAA |
55 (24.6) |
21 (36.2) |
|
CE |
130 (58.0) |
25 (43.1) |
|
Others |
39 (17.4) |
12 (20.7) |
|
Previous stroke, % (SD) |
49 (21.9) |
10 (17.2) |
0.44* |
HTN, % (SD) |
121 (54.0) |
38 (65.5) |
0.12* |
DM, % (SD) |
62 (27.7) |
15 (25.9) |
0.78* |
HL, % (SD) |
33 (14.7) |
7 (12.1) |
0.60* |
Current smoking, % (SD) |
31 (13.8) |
7 (12.1) |
0.73* |
Atrial fibrillation, % (SD) |
120 (53.6) |
25 (43.1) |
0.16* |
Previous use of antithrombotics, % (SD) |
87 (38.8) |
18 (31.0) |
0.27* |
Previous use of statin, % (SD) |
49 (21.9) |
5 (8.6) |
0.02* |
WBC, x1000/µL (SD) |
8.70 (3.26) |
9.46 (4.42) |
0.03† |
Creatinine, mg/dL (SD) |
0.98 (0.60) |
0.87 (0.30) |
0.36† |
Hemoglobin, g/dL (SD) |
13.7 (2.1) |
13.4 (2.2) |
0.76† |
Platelet, x1000/µL (SD) |
221.9 (70.9) |
250.9 (125.9) |
0.02† |
LDLc, mg/dL (SD) |
46.1 (30.8) |
48.2 (34.8) |
0.03† |
HbA1c, % (SD) |
6.2 (1.4) |
5.9 (0.7) |
<0.001† |
Prothrombin time, INR (SD) |
1.08 (0.25) |
1.05 (0.11) |
0.22† |
CRP, mg/dL (SD) |
12.2 (25.0) |
14.0 (26.2) |
0.28† |
Initial random glucose, mg/dL (SD) |
141.2 (51.6) |
148.9 (63.7) |
0.95† |
SBP, mmHg (SD) |
149.2 (26.4) |
149.3 (25.9) |
0.70† |
Table 2. Intervention-related findings and outcomes according to reperfusion status
Successful reperfusion (n=224) |
Unsuccessful reperfusion (n=58) |
p-value |
|
Reperfusion therapy, % (SD) |
|
0.67* |
|
EVT only |
105 (46.9) |
29 (50.0) |
|
Combined IVT and EVT |
119 (53.1) |
29 (50.0) |
|
LVO location, % (SD) |
0.76* |
||
ICA |
17 (7.6) |
3 (5.2) |
|
M1/M2 MCA |
183 (81.7) |
46 (79.3) |
|
BA/VA |
24 (10.7) |
9 (15.5) |
|
Collateral status, % (SD) |
0.06* |
||
poor |
100 (44.6) |
22 (37.9) |
|
intermediate |
35 (15.6) |
17 (29.3) |
|
good |
89 (39.7) |
19 (32.8) |
|
NLR (IQR) |
2.48 (1.52-4.87) |
3.92 (2.29-8.14) |
0.002† |
PLR (IQR) |
113.2 (80.9-168.8) |
147.2 (109.0-227.9) |
0.001† |
Infarct volume, cm3 (IQR) |
12.25 (2.64-50.78) |
28.88 (9.20-102.63) |
<0.001† |
- Results – line 145-147: “The unsuccessful reperfusion group likely had lower pre-stroke statin level, higher white blood cell count and low-density lipoprotein level, and lower HbA1c level than the successful reperfusion group (Table 1).” The statistically difference between the group are they have any clinical relevance? The numbers are actually in the normal range??
Response: Thank you for your comment. We cautiously speculate that the prior use of statins could affect reperfusion status. However, this has not been proven yet and we plan to collect data and study this issue soon. As you mentioned, the since laboratory results (WBC, HbA1c and LDLc) were all within the normal range, the differences seem to be of little clinical relevance. We have revised the Results as follows:
[Results] [page 10, paragraph 1]
We analyzed 3,169 consecutive patients with acute ischemic stroke. Among these patients, 606 underwent IVT and EVT. Of the 606 patients, 282 who underwent EVT were included in our study. Of the 282 patients, 58 (20.6%) were in the unsuccessful reperfusion group. The unsuccessful reperfusion group likely had fewer prior statin users than the successful reperfusion group. Some laboratory results (white blood cell count, low-density lipoprotein level, and HbA1c level) showed statistically but not clinically significant differences between the two groups (Table 1).
- In the unsuccessful reperfusion group had a significant increase of infarct volume compare to the successful group. Maybe this finding can explain the effect of the inflammatory response and the high NLR in this group?
Response: We appreciate your comment. Several lines of evidence have showed that inflammation initiates the breakdown of the BBB. Inflammatory molecules subsequently invade the brain parenchyma, thereby aggravating ischemic injury. In addition, neutrophils, which are activated in the intravascular walls at an early stage, can promote intravascular clogging and destabilize the BBB. Hence, we assumed that a higher NLR, as a surrogate marker for acute inflammatory response, could be associated with infarct volume. In addition, our results showing the association between higher NLR and unsuccessful reperfusion corroborate the finding of increasing infarct volume in the unsuccessful group. We described this assumption in the Discussion section.
[Discussion] [page 12, paragraph 2]
Although this intravascular inflammatory reaction occurs, inflammation also initiates the breakdown of the blood-brain barrier (BBB), and cytokines and adhesion molecules invade the brain parenchyma, thereby further enhancing ischemic damage.28-30 In the pathogenesis of inflammatory cascades, neutrophils and platelets play a key role in inflammation and thrombosis. Several evidences have shown that neutrophils, which are activated early in intravascular adhesion, contribute to postischemic injury by intravascular clogging, destabilizing BBB by matrix proteinases, and generating reactive oxygen species.31-32 In addition, activated platelets contribute to plaque destabilization by promoting thrombi and inducing inflammatory response.33 This pathological mechanism underlying the inflammatory reaction after acute ischemic stroke may explain the potential associations between high NLR and PLR and both reperfusion rate and infarct volume after EVT. Accordingly, these pathological findings support our main results.
- The authors suggest (line 228): “In the trend of increasing performance of EVT according to recent stroke guidelines, our study could raise the interest of application of NLR and PLR to predict outcomes in real-world practice of EVT”. It is unclear how physicians can use these finding to inform patient about his prognosis and who this finding will change the clinical practice for the individual patient.
Response: We appreciate your comment. Our aim in this study was to evaluate the predictive role of NLR and PLR in acute stroke management. We aim to show in our paper that the joint consideration of NLR/PLR, as well as other powerful predictors for reperfusion status (such as stent retriever device, characteristics of thrombus, and operator factors), could be help predict the reperfusion status by blood sampling early after admission. In addition, NLR/PLR could be useful markers of poor prognosis in patients with futile reperfusion. We have toned down our description as follows:
[Discussion] [page 13, paragraph 1]
Previous studies that limited subjects receiving EVT have shown that NLR and PLR could be predictors for hemorrhagic transformation, functional outcome, and mortality.12, 21-22, 34 Our novel findings also showed that the different predictive values of NLR and PLR for reperfusion state broadened the choice of biomarkers to establish the EVT strategy. We tentatively propose that, although there are other strong predictors for successful reperfusion, joint consideration of NLR/PLR may help predict reperfusion status by blood sampling early after admission.
- A more detailed explanation is needed for this finding “PLR was not associated with an increased final infarct volume in subjects with anterior circulation infarction”??
Response: We appreciate your opinion. We speculate that the main reason for the lack of an association between PLR and infarct volume is the small sample size of patients with anterior circulation infarct. Another explanation may be subtle differences between what NLR and PLR represent. The NLR mainly represents inflammatory injury, whereas the PLR represents thrombus formation in addition to plaque inflammation, especially under atherosclerotic conditions. The majority of our patients with anterior circulation were found to have cardioembolism (LAA 28.5% and CE 55.0%). The low rates of LAA may explain the attenuated association between PLR and infarct volume in anterior circulation infarction. We have revised the Discussion section as follows:
[Discussion] [page 14, paragraph 1]
In addition, PLR plays a key role in thrombus and inflamed plaque formation, especially under atherosclerotic conditions.36 The majority of our patients with anterior circulation infarction were found to have CE (LAA: 28.5% and CE: 55.0%). The low rate of LAA could explain the attenuated association between PLR and infarct volume in anterior circulation infarction. Further studies with large sample sizes and advanced programs for precise calculation of infarct size are warranted to generalize our results.
- It is a two center retrospective study and not a multi-center study
Response: We appreciate your opinion. We have revised as follows:
[Discussion] [page 15, paragraph 2]
Despite the strengths of this study, including its two-consecutive registry database, this study has several limitations.

Reviewer 2 Report
The study highlights the role of NLR and PLR in predicting unsuccessful reperfusion after EVT. The results are interesting and the text is well-organized and well-written with logical formatting.
I have the following comments.
1. There are various factors affecting successful reperfusion after EVT including EVT technique (eg. Stent retriever, contact aspiration, etc), presence of intracranial atherosclerotic disease, and location of LVO which were not described in this study. It would be better to add these variables in Table 1.
2. Table 1: The NLR does not seem normally distributed. Please assess the normality of distribution of NLR and PLR and adapt the statistical analysis if not normal distribution.
3. Line 143-145. Please consider the use of a flow diagram.
4. Line 180-182. The statistical method regarding sensitivity analysis done with patients with only anterior circulation infarction was not described in the method section. Please add the statistical analysis plan in the method section.
5. Discussion section: Please present deep mechanistic interpretations of the findings that both high NLR and PLR were associated with lower reperfusion rate.
Author Response
- 1. There are various factors affecting successful reperfusion after EVT including EVT technique (eg. Stent retriever, contact aspiration, etc), presence of intracranial atherosclerotic disease, and location of LVO which were not described in this study. It would be better to add these variables in Table 1.
Response: We appreciate your comments. Technique and individual operator factors at each center were not available in our database. Regarding EVT procedures, the choice of device and intervention strategy were at the discretion of the interventionists at each institution, according to recent guidelines. We have described this limitation in the revised Discussion section. The presence of intracranial atherosclerotic disease was defined as LAA and presented in Table 1. As per the reviewer’s request, we have added the location of LVO in Table 2. The revisions made are as follows:
[Methods] [page 7, paragraph 4]
The primary outcome measure was poor reperfusion rate defined by modified Thrombolysis in Cerebral Infarction (TICI) grades 0 to 2a.25 Vascular neurologists in each institution made decisions on whether to perform reperfusion therapy. Regarding EVT procedures, the choice of device and intervention strategy was made at the discretion of the interventionists at each institution, according to recent guidelines.
[Discussion] [page 15, paragraph 2]
Third, although we adjusted for several variables that may affect the outcomes, unmeasured confounding factors could hinder the applicability of our main findings. The outcomes after EVT can be influenced by procedure-related factors, including the type of stent retriever, operator technique and expertise, and procedure time, which were not available for analysis in our study. Fourth, although the aim of this study was to evaluate the impact of NLR and PLR on outcome, other inflammatory markers that arise during the acute stroke stage were not available in this study.
[Table 2]
Table 2. Interventional findings and outcomes according to reperfusion status
Successful reperfusion (n=224) |
Unsuccessful reperfusion (n=58) |
p-value |
|
Reperfusion therapy, % (SD) |
|
0.67* |
|
EVT only |
105 (46.9) |
29 (50.0) |
|
Combined IVT and EVT |
119 (53.1) |
29 (50.0) |
|
LVO location, % (SD) |
0.76* |
||
ICA |
17 (7.6) |
3 (5.2) |
|
M1/M2 MCA |
183 (81.7) |
46 (79.3) |
|
BA/VA |
24 (10.7) |
9 (15.5) |
|
Collateral status, % (SD) |
0.06* |
||
poor |
100 (44.6) |
22 (37.9) |
|
intermediate |
35 (15.6) |
17 (29.3) |
|
good |
89 (39.7) |
19 (32.8) |
|
NLR (IQR) |
2.48 (1.52-4.87) |
3.92 (2.29-8.14) |
0.002† |
PLR (IQR) |
113.2 (80.9-168.8) |
147.2 (109.0-227.9) |
0.001† |
Infarct volume, cm3 (IQR) |
12.25 (2.64-50.78) |
28.88 (9.20-102.63) |
<0.001† |
Abbreviation: SD, standard deviation; EVT, endovascular treatment; IVT, intravenous thrombolysis; LVO, larger vessel occlusion; ICA, internal carotid artery; MCA, middle cerebral artery; BA, basilar artery; VA, vertebral artery; NLR, neutrophil/lymphocyte ratio; PLR, platelet/lymphocyte ratio; IQR, interquartile range
- Table 1: The NLR does not seem normally distributed. Please assess the normality of distribution of NLR and PLR and adapt the statistical analysis if not normal distribution.
Response: We appreciate your comment. Since both NLR and PLR were not normally distributed, the statistical method used to analyze them was changed as follows:
[Table 2]
Table 2. Interventional findings and outcomes according to reperfusion status
Successful reperfusion (n=224) |
Unsuccessful reperfusion (n=58) |
p-value |
|
Reperfusion therapy, % (SD) |
|
0.67* |
|
EVT only |
105 (46.9) |
29 (50.0) |
|
Combined IVT and EVT |
119 (53.1) |
29 (50.0) |
|
LVO location, % (SD) |
0.76* |
||
ICA |
17 (7.6) |
3 (5.2) |
|
M1/M2 MCA |
183 (81.7) |
46 (79.3) |
|
BA/VA |
24 (10.7) |
9 (15.5) |
|
Collateral status, % (SD) |
0.06* |
||
poor |
100 (44.6) |
22 (37.9) |
|
intermediate |
35 (15.6) |
17 (29.3) |
|
good |
89 (39.7) |
19 (32.8) |
|
NLR (IQR) |
2.48 (1.52-4.87) |
3.92 (2.29-8.14) |
0.002† |
PLR (IQR) |
113.2 (80.9-168.8) |
147.2 (109.0-227.9) |
0.001† |
Infarct volume, cm3 (IQR) |
12.25 (2.64-50.78) |
28.88 (9.20-102.63) |
<0.001† |
Abbreviation: SD, standard deviation; EVT, endovascular treatment; IVT, intravenous thrombolysis; LVO, larger vessel occlusion; ICA, internal carotid artery; MCA, middle cerebral artery; BA, basilar artery; VA, vertebral artery; NLR, neutrophil/lymphocyte ratio; PLR, platelet/lymphocyte ratio
* Calculated using the chi-square test
† Calculated using the Mann-Whitney U test
- Line 143-145. Please consider the use of a flow diagram.
Response: We appreciate your comment. As per the reviewer’s request, we have added a flow chart as Supplementary Figure 1 as follows:
[Supplementary Figure 1]
- Line 180-182. The statistical method regarding sensitivity analysis done with patients with only anterior circulation infarction was not described in the method section. Please add the statistical analysis plan in the method section.
Response: We appreciate your comment. We have revised the Method section as follows:
[Methods] [page 8, paragraph 2]
A sensitivity analysis using linear regression was performed to evaluate whether both NLR and PLR could be associated with infarct volume in anterior circulation infarctions only.
Subgroup analysis was performed to evaluate the impact of NLR and PLR on unsuccessful reperfusion according to stroke mechanisms. We separately performed binary logistic regression analysis in patients with stroke with large artery atherosclerosis (LAA) and cardioembolism (CE) using the aforementioned statistical methods.
- Discussion section: Please present deep mechanistic interpretations of the findings that both high NLR and PLR were associated with lower reperfusion rate.
Response: Thank you for your comment. As per the reviewer’s comment, we have revised the Discussion section as follows:
[Discussion] [page 12, paragraph 2]
After vessel occlusion in acute stroke, the inflammatory cascade is activated on the vessel endothelium and platelets immediately.26-27 The intravascular endothelium and platelet are stressed by stagnant blood flow and induced release of P-selectin in the endothelium and platelet. P-selectin could bind to circulating leukocytes, facilitating cluster formation of leukocytes, causing intravascular clogging on the endothelial surface. Neutrophils and platelets play a key role in the inflammation and thrombosis occurring subsequent to inflammatory cascades. The NLR and PLR may be novel indicators of the intensity of systemic inflammation, which may allow quantification of these inflammatory reactions. Higher PLR in particular has been associated with thrombus and inflamed intravascular plaque formation. In addition, activated platelets contribute to plaque destabilization by promoting thrombi and inducing an inflammatory response.28 Therefore, we can speculate that higher NLR and PLR may be associated with an increased risk of unsuccessful reperfusion.

Reviewer 3 Report
This manuscript by Lee et al. focused on the association between neutrophil-to-lymphocyte ratio (NLR) or platelet-to-lymphocyte ratio (PLR) and reperfusion rate or prognosis after endovascular treatment (EVT) in patients with 3,169 patients with acute ischemic stroke. As authors mentioned, inflammation is a key mechanism of atherosclerosis in patients stroke as well as other vascular disease. Therefore, the concept of this manuscript in understandable and results seems reasonable. Although this manuscript seems written well, authors may want to consider several issues as follows.
Major comments:
1) No major concern to be resolved can be found.
Minor comments:
1) In title, it is unknown whether subjects of this study were patients with acute ischemic stroke or other vascular diseases. I imagined large aortic diseases from this title.
2) In Table 1, LDL may be LDL cholesterol.
3) In Figure 1, Tables 2 and 3, ranges of tertiles for NLR T1 to T3 and PLR T1 to T3 should be shown.
Author Response
Reviewer#3
Major comments:
1) No major concern to be resolved can be found.
Minor comments:
1) In title, it is unknown whether subjects of this study were patients with acute ischemic stroke or other vascular diseases. I imagined large aortic diseases from this title.
Response: We appreciate your comment. We have revised the Title as follows:
Title: The Neutrophil-to-Lymphocyte and Platelet-to-Lymphocyte Ratios Predict Reperfusion and Prognosis after Endovascular Treatment of Acute Ischemic Stroke
2) In Table 1, LDL may be LDL cholesterol.
Response: We appreciate your comment. We have revised as follows:
Successful reperfusion (n=224) |
Unsuccessful reperfusion (n=58) |
p-value |
|
Age (SD) |
68.7 (13.7) |
72.8 (11.7) |
0.17 |
Male, % (SD) |
135 (60.3) |
27 (46.6) |
0.07 |
BMI (SD) |
25.5 (23.2) |
23.4 (3.8) |
0.59 |
NIHSS (IQR) |
14 (9-18) |
15 (11-18) |
0.13 |
Interval from arrival to puncture, min (IQR) |
96.5 (75.0-134.0) |
101.5 (73.0-156.0) |
0.18 |
Stroke mechanism, % (SD) |
0.12 |
||
LAA |
55 (24.6) |
21 (36.2) |
|
CE |
130 (58.0) |
25 (43.1) |
|
Others |
39 (17.4) |
12 (20.7) |
|
Previous stroke, % (SD) |
49 (21.9) |
10 (17.2) |
0.44 |
HTN, % (SD) |
121 (54.0) |
38 (65.5) |
0.12 |
DM, % (SD) |
62 (27.7) |
15 (25.9) |
0.78 |
HL, % (SD) |
33 (14.7) |
7 (12.1) |
0.6 |
Current smoking, % (SD) |
31 (13.8) |
7 (12.1) |
0.73 |
Atrial fibrillation, % (SD) |
120 (53.6) |
25 (43.1) |
0.16 |
Previous use of antithrombotics, % (SD) |
87 (38.8) |
18 (31.0) |
0.27 |
Previous use of statin, % (SD) |
49 (21.9) |
5 (8.6) |
0.02 |
WBC, x1000/µL (SD) |
8.70 (3.26) |
9.46 (4.42) |
0.03 |
Creatinine, mg/dL (SD) |
0.98 (0.60) |
0.87 (0.30) |
0.36 |
Hemoglobin, g/dL (SD) |
13.7 (2.1) |
13.4 (2.2) |
0.76 |
Platelet, x1000/µL (SD) |
221.9 (70.9) |
250.9 (125.9) |
0.02 |
LDLc, mg/dL (SD) |
46.1 (30.8) |
48.2 (34.8) |
0.03 |
HbA1c, % (SD) |
6.2 (1.4) |
5.9 (0.7) |
<0.001 |
Prothrombin time, INR (SD) |
1.08 (0.25) |
1.05 (0.11) |
0.22 |
CRP, mg/dL (SD) |
12.2 (25.0) |
14.0 (26.2) |
0.28 |
Initial random glucose, mg/dL (SD) |
141.2 (51.6) |
148.9 (63.7) |
0.95 |
SBP, mmHg (SD) |
149.2 (26.4) |
149.3 (25.9) |
0.7 |
Abbreviation: SD, standard deviation; BMI, body mass index; NIHSS, National Institute Health of Stroke Scale; IQR, interquartile range; LAA, large artery atherosclerosis; CE, cardioembolism; HTN, hypertension; DM, diabetes mellitus; HL, hyperlipidemia; EVT, endovascular treatment; IVT, intravenous thrombolysis; WBC, white blood cell; LDLc, low density lipoprotein cholesterol; HbA1c, glycated hemoglobin; INR, international normalized ratio; CRP, C reactive protein; SBP, systolic blood pressure
3) In Figure 1, Tables 2 and 3, ranges of tertiles for NLR T1 to T3 and PLR T1 to T3 should be shown.
Response: We appreciate your comment. We have revised the Figure legend and Table captions as follows:
[Figure legends]
Figure legends
Figure 1. Distribution of final infarct volume according to tertile of NLR and PLR
Abbreviation: NLR, neutrophil to lymphocyte ratio; PLR, platelet to lymphocyte ratio; T1, lowest tertile; T2, middle tertile; T3, highest tertile
Range of tertiles: NLR T1 (0.30-1.88), NLR T2 (1.90-4.15), NLR T3 (4.15-54.75); PLR T1 (24.5-94.1), PLR T2 (94.2-157.2), PLR T3 (158.6-754.4)
[Table]
Table 3. logistic regression analysis showing effect of NLR and PLR on unsuccessful reperfusion
aOR |
95% CI |
p-value |
|
aOR |
95% CI |
p-value |
|
raw NLR |
1.11 |
1.04-1.19 |
0.001 |
raw PLR |
1.004 |
1.001-1.01 |
0.01 |
NLR T1 |
reference |
PLR T1 |
reference |
||||
NLR T2 |
1.33 |
0.56-3.12 |
0.52 |
PLR T2 |
2.56 |
1.06-6.22 |
0.04 |
NLR T3 |
2.51 |
1.11-5.70 |
0.03 |
PLR T3 |
2.84 |
1.15-6.99 |
0.02 |
Abbreviation: NLR, neutrophil/lymphocyte ratio; PLR, platelet/lymphocyte ratio; aOR, adjusted odds ratio; CI, confidence interval; T1, lowest tertile; T2, middle tertile; T3, highest tertile
Range of tertiles: NLR T1 (0.30-1.88), NLR T2 (1.90-4.15), NLR T3 (4.15-54.75); PLR T1 (24.5-94.1), PLR T2 (94.2-157.2), PLR T3 (158.6-754.4)
Table 4. logistic regression analysis showing effect of NLR and PLR on poor functional outcome (mRS 3 to 6)
aOR |
95% CI |
p-value |
|
aOR |
95% CI |
p-value |
|
raw NLR |
1.20 |
1.06-1.35 |
0.004 |
raw PLR |
1.01 |
1.004-1.02 |
<0.001 |
NLR T1 |
reference |
PLR T1 |
reference |
||||
NLR T2 |
2.32 |
1.15-4.70 |
0.02 |
PLR T2 |
1.52 |
0.75-3.09 |
0.24 |
NLR T3 |
3.67 |
1.67-8.06 |
0.001 |
PLR T3 |
2.59 |
1.10-5.55 |
0.02 |
Abbreviation: NLR, neutrophil/lymphocyte ratio; PLR, platelet/lymphocyte ratio; mRS; modified Rankin Scale; aOR, adjusted odds ratio; CI, confidence interval; T1, lowest tertile; T2, middle tertile; T3, highest tertile
Range of tertiles: NLR T1 (0.30-1.88), NLR T2 (1.90-4.15), NLR T3 (4.15-54.75); PLR T1 (24.5-94.1), PLR T2 (94.2-157.2), PLR T3 (158.6-754.4)

Round 2
Reviewer 1 Report
no comment
Reviewer 2 Report
The authors have thoughtfully addressed the reviewer's questions/comments and made the appropriate updates.
I have no further comments. Thank you.